# Everything in Moderation: Lessons Learned by Exploiting Moderate Replication Stress in Cancer

**DOI:** 10.3390/cancers11091320

**Published:** 2019-09-06

**Authors:** Deborah Nazareth, Mathew J. K. Jones, Brian Gabrielli

**Affiliations:** 1Mater Research Institute, The University of Queensland, Translational Research Institute, Brisbane, QLD 4102, Australia; dnazareth@cmri.org.au; 2Molecular Biology Program, Sloan Kettering Institute, Memorial Sloan Kettering Cancer Center, New York, NY 10065, USA; mathew.jones@uq.edu.au; 3The University of Queensland Diamantina Institute, The University of Queensland, Translational Research Institute, Brisbane, QLD 4102, Australia

**Keywords:** replication stress, CHK1 inhibitor, immunotherapy

## Abstract

The poor selectivity of standard cytotoxic chemotherapy regimens causes severe side-effects in patients and reduces the quality of life during treatment. Targeting cancer-specific vulnerabilities can improve response rates, increase overall survival and limit toxic side effects in patients. Oncogene-induced replication stress serves as a tumour specific vulnerability and rationale for the clinical development of inhibitors targeting the DNA damage response (DDR) kinases (CHK1, ATR, ATM and WEE1). CHK1 inhibitors (CHK1i) have served as the pilot compounds in this class and their efficacy in clinical trials as single agents has been disappointing. Initial attempts to combine CHK1i with chemotherapies agents that enhance replication stress (such as gemcitabine) were reported to be excessively toxic. More recently, it has emerged that combining CHK1i with subclinical doses of replication stress inducers is more effective, better tolerated and more compatible with immunotherapies. Here we focus on the lessons learned during the clinical development of CHK1i with the goal of improving the design of future clinical trials utilizing DDR inhibitors to target replication stress in cancer.

## 1. Introduction

Many conventional cytotoxic chemotherapies suppress the proliferation of cancer cells by blocking DNA synthesis via inhibiting nucleotide production (e.g., gemcitabine/Gemzar and 5-fluorouracil/Adrucil) or generating inter-strand DNA crosslinks (e.g., platinum-based compounds Platinol and Oxalipatin). Appropriately administering chemotherapy is extremely challenging as many chemotherapy agents have a narrow therapeutic window separating the minimal effective dose and the maximum tolerated dose, which is made even more difficult by pharmacokinetic variability among cancer patients. Failure to optimally dose chemotherapy causes toxic side-effects in many highly proliferative healthy tissues (e.g., hair follicles, gastrointestinal tract and bone marrow). Chemotherapies can also be toxic to rapidly dividing immune cells providing a significant barrier to harnessing the immune system during treatment regimens that combine chemotherapy with immune checkpoint inhibition. Hence the design and dosing of chemotherapy regimens is crucial for maximizing tumour eradication.

The poor tumour selectivity of conventional chemotherapies was a major justification for large investments made in cancer genomics, which over the past decade has provided a catalogue of oncogenic driver mutations and signaling pathways that has guided the development of therapeutic interventions. Many targeted cancer treatments have now emerged that focus on tumour specific drivers (e.g., BRAF inhibitors to combat mutant BRAF-dependent melanomas and antibody-based therapies to address over-expression or aberrant EGFR signaling). The advantage of these targeted approaches is the relative ease of identification of the patients likely to benefit, i.e., patients with the mutation or amplification, while a major shortcoming is the reliance on the presence of a single defect which allows solid cancers in particular to rapidly develop resistance to these targeted agents [1].

An alternative approach relies on targeting a pathway that is essential for the viability of cells with particular defects, without directly targeting the defects themselves. These we have termed pathways of dependence. Selectively inhibiting these pathways in cells with the defect destroys them whereas normal cells that are not reliant on the inhibited pathway are relatively unaffected. This review will focus on strategies to target replication stress, a defect commonly found in cancers, particularly aggressive cancers such a melanomas and lung cancers. A number of targets have been identified to selectively target tumours with increased replication stress, either endogenous or exogenously applied. These have focused primarily on the S phase cell cycle checkpoint kinases, CHK1, ATR and WEE1. However, the success of compounds targeting these kinases as single agents has been quite modest. Here we will discuss approaches to improve targeting of tumours with elevated replication stress while minimizing normal tissue toxicity, particularly of the innate and adaptive immune system.

## 2. Cellular Responses to Replication Stress

High fidelity genome replication is a major barrier to carcinogenesis [2]. The sustained proliferative signaling caused by activated oncogenes places an enormous burden on the replication machinery, which can potentially reduce the fidelity of replication. Depending on the degree of replication stress, individual forks respond differently. For example, during moderate replication stress, elongating forks predominately slow (e.g., 1.5–2.0 kb/min to 1.0–0.5 kb/min) compared to the complete stalling and collapse of replication forks into DSBs during high levels of replication stress [3]. The outcomes on DNA replication and the cellular mechanisms responsible are distinct although some of the components are common to both.

Replication stress can be the result of either endogenous or exogenous stressors. Moderate stress can be the result of reduced dNTP pools through reduced nutrient levels or a low O_2_ environment, as the RRM2 subunit of ribonucleotide reductase is particularly sensitive to O_2_ levels [4]. Under conditions of low dNTP, the level of RRM2 is increased by a combination of increased E2F1-dependent transcription and increased protein stability [5,6,7,8]. Endogenous stresses include complex DNA structures and tightly bound proteins which slow down the replication fork progression. Exogenous stresses include ultraviolet radiation and many of the chemotherapeutic drugs that inhibit dNTP production or promote DNA damage and crosslinks that can block replication fork progression. Cellular transformation by oncogenes also promotes replications stress [9].

High level replication stress could be fatal to a cell, thus mechanisms that repair or overcome the damage incurred by incomplete or reduced fidelity of replication are critical for the continued viability of the cell. The S phase cell cycle checkpoint is triggered by high level replication stress that blocks fork progression, thereby halting cell cycle progression until the fault is repaired. This ensures complete replication of the genome before allowing the cell to progress into mitosis. The S phase checkpoint is triggered by the replication fork encountering conditions that are not permissive for continued fork progression, such as insufficient dNTPs, bulky DNA adducts, inter-strand DNA crosslinks or tightly bound protein complexes [10]. The MCM helicase complex continues to unwind the DNA duplex while the polymerase complex is arrested creating long stretches of single stranded DNA (ssDNA) in the process. The replication protein A (RPA) complex binds ssDNA and recruits and activates the apical cell cycle checkpoint signaling kinase Ataxia telangiectasia and RAD3-related (ATR), through the colocalization of other factors including ATRIP, TopBP1 and ETAA1 [10]. ATR activation triggers S phase arrest through its activation of the effector kinase, Checkpoint kinase 1 (CHK1) [11,12] (Figure 1). The role of ATR-CHK1 activation in this scenario is stabilization of the stalled replication fork. Both ATR and CHK1 protect the replication fork by preventing excessive single strand DNA formation, fork remodeling and fork cleavage. Importantly, checkpoint activation also prevents the firing of late origins [13], thereby effectively inhibiting all DNA replication until the blocked forks are resolved. This is through CHK1-mediated degradation of CDC25A which prevents the activation of the S phase CDK2/cyclin complexes and hence blocks cell cycle progression [14,15,16]. WEE1 is the negative regulator of these CDK2 complexes, directly opposing CDC25A by phosphorylating the inhibitory CDK2 Ty15 residue dephosphorylated by CD25A, and inhibition of WEE1 is sufficient to promote unscheduled S phase entry and replication [17] (Figure 1A).

Moderate replication stress is triggered by conditions that produce slowing of replication forks, such as suboptimal levels of dNTPs. This triggers a mechanism involving the initiation of replication from normally dormant replication origins within active replication factories to ensure replication continues [18]. Under these conditions there is slowing of DNA replication and progression through S phase but it is not completely stalled [18,19]. It is unclear as to whether ATR is activated, but CHK1 is activated to block late replication origin firing [18,19,20]. The outcomes of CHK1 activation are clearly quite different: continued DNA replication and S phase progression in the presence of moderate stress, and complete arrest in the face of high level stress (Figure 1B).

Increased replication stress is one of the major outcomes of treatment with conventional chemotherapeutic drugs such as ribonucleotide reductase inhibitors (hydroxyurea and Gemcitabine), nucleotide analogues (Fludarabine), DNA crosslinkers and alkylating agents (Cisplatin and mitomycin C), and Topoisomerase I inhibitors (Camptothecin) [21]. These drugs are used in the clinic at their maximum tolerated dose, imposing a strong S phase arrest which can lead to fork collapse and DNA DSBs. In cancer, these cells either die after prolonged S phase arrest or the S phase checkpoint arrest can fail and cells resume progressing through mitosis without repairing the initial lesions [12]. This is the basis of the cytotoxicity of these agents. However, these drugs are relatively unselective and produce this high level replication stress in all cells progressing through S phase, including normal cells. This is the basis of the dose limiting toxicity to normal tissue, commonly leukopenia due to blocking proliferation of rapidly dividing haemopoietic lineages and other rapidly proliferating cell types such as in the intestinal crypts, thereby reducing their clinical efficacy [21]. In contrast to the chemotherapy-imposed high level replication stress, a low level of replication stress is a common feature of a high proportion of cancers, although this is not sufficient to delay cell cycle progression. Thus, selective targeting of the low level endogenous replication stress should provide the tumour selectivity desired in an effective targeted therapy.

## 3. Selective Targeting of Elevated Replication Stress

Inhibitors of ATR-CHK1-WEE1 have been shown to effectively target cells with increased replication stress. These drugs were originally developed as chemosensitizers to ablate the S and G2 phase cell cycle checkpoint arrest that is a common outcome of chemotherapy treatment. The checkpoint arrest reduces the effectiveness of the chemotherapies by promoting repair of the DNA damage inflicted by the drugs [22]. Combination with the checkpoint signaling inhibitors increases DNA damage through failure to repair the original DNA lesions, and in many cases promoting more DNA damage leading to cell death [23].

CHK1i are chemosensitizers to a broad range of drugs that trigger S and G2 phase cell cycle checkpoint arrest [22], their cytotoxicity appears to be partially dependent on p53 functional status and the type of DNA damaging agent used [24]. CHK1i have limited single agent activity in vitro or in vivo [25,26,27,28], and only in cell lines with high levels of endogenous replication stress [26,29,30]. The CHK1i hypersensitive melanoma cell lines have lost their ability to S phase checkpoint arrest as an adaptation to the high levels of endogenous replication [26]. However, the high level endogenous stress does not increase activated CHK1 levels [30].

CHK1i synergise most strongly with chemotherapeutic drugs that impose replication stress [24]. In vitro and in vivo, CHK1i synergise strongly with gemcitabine and HU, reducing the effective dose of these replication stress inducers by at least 10-fold [19,24,31,32,33]. When used in combination with drugs that promote strong replication stress and S phase arrest, the consequence is excessive replication fork firing and RPA exhaustion [7,11], leading to MUS81-EME1/2-dependent DNA damage and cell death [34] (Figure 2A). The DNA damage and death can be blocked by inhibiting CDK2 and CDC7-dependent helicase activation [25,35].

This combination has not translated in clinical trials where only low response rates were observed. These trials have all used gemcitabine as the replication stress inducer at the standard of care monotherapy dose of 500–1000 mg/m^2^ dose [36,37,38]. This dose produces grade 3–4 toxicities, normally hematological toxicities, vomiting and diarrhea in 10–20% of patients [39], and triggers S phase arrest in patient tumours and proliferating normal tissue, indicative of high level replication stress [32]. Thus, it is not surprising that higher levels of toxicities were observed when combined with a CHK1i inhibitor that sensitized the patient to gemcitabine. WEE1i and ATRi have also been shown to effectively promote cell killing in combination with high replication inducers [7,40], demonstrating that inhibition of the S phase checkpoint is sufficient to promote cell killing under these conditions.

In contrast to high replication stress, moderate replication stress induced with low doses (<0.2 mM) of hydroxyurea (HU) and (<30 nM) gemcitabine that slows replication and triggers dormant origin activation is insensitive to either ATRi or WEE1i, but CHK1i treatment produces similar levels of RPA exhaustion, DNA damage and tumour cell death as found with high replication stress [7,19,31,40] (Figure 2B). This indicates that ATR and WEE1 are not centrally involved in the moderate replication stress response, whereas CHK1 is essential to this response. Importantly, moderate replication stress produced much lower levels of normal tissue toxicity in vitro and in vivo, especially in the hematopoietic cell linages that constitute the innate and adaptive immune systems [19].

These data suggest that to obtain maximum anti-tumour activity with minimal normal tissue toxicity, the level of replication stress inducer used is critical (Figure 3). Proliferating normal tissue which has little or no replication stress is relatively unaffected by low dose of HU or gemcitabine that do not impose an S phase arrest, but slows replication and triggers dormant origin firing. When combined with CHK1i (neither ATRi or WEE1i are effective under these conditions), the level of RPA exhaustion or DNA damage is either insufficient to trigger cell death or with removal of drugs cells repair is affected and cells survive and continue to proliferate [19]. However, tumour cells with already elevated replication stress can be utilizing three-fold more origins than normal cells [41] suggesting that rate limiting replication factors may be almost fully utilized, and activation of further dormant origins by imposing moderate replication pushes these to critical. Inhibition of CHK1 and consequent late origin firing results in the massive DNA damage observed. By contrast, high level stress can readily exhaust critical replication factors by inhibiting all replication forks in normal and tumour cells, so that when combined with either ATRi, CHK1i or WEE1i activation of late firing origins results in the DNA damage and cell death in both normal and tumour tissue, and the consequent excessive normal tissue toxicity (Figure 3).

## 4. Choice of Replication Stress Inducer

The model in Figure 2 indicates that choice of replication stress inducer and dose used are critical to effectively target the tumour while retaining normal tissue viability and proliferative potential. While many chemotherapies promote replication stress, our model suggests that those promoting high level replication stress by imposing an S phase arrest are likely to have high levels of normal tissue toxicity and disable the immune responses. It is possible to titrate HU and gemcitabine to levels where they have little effect on S phase but synergise with CHK1i to kill tumour cells [19,40]. While this is possible in vitro, in patients it is likely to be very challenging with gemcitabine as it is delivered intravenously, and concentrations as low as 50 nM produce an S phase arrest [19,32,40,42]. Thus, there is a small therapeutic window likely due to the irreversible inhibition of ribonucleotide reductase, and incorporation of chain terminating gemcitabine metabolites into the DNA, both of which contribute to gemcitabine’s mechanisms of action [43]. By contrast, HU is an orally active reversible inhibitor of ribonucleotide reductase that can readily achieve plasma concentrations of 1–2 mM without significant toxicity [44], indicating that continuous low dosing with the HU should be possible. Other replication stress inducers such as TOPOI inhibitors and cisplatin poorly sensitize to CHK1i [24,31]. PARP inhibitors (PARPi) can trap PARP1 on the chromatin and this is responsible for the rapid appearance of DNA damage in S phase cells, a consequence of replication forks encountering the trapped PARP1 and stalling [45,46]. PARPi treatment induced a modest reduction in S phase progression suggesting that it produces moderate replication stress [45,47]. However, PARPi have been reported to sensitize the cells to ATRi, CHK1i and WEE1i [47,48,49,50,51,52]. This may be due to the different PARPi trapping affinity produced by the different PARPi [46], with the lower affinity trapping PARPi producing moderate replication stress and high affinity trappers high replication stress.

Other combinations that target replication stress include ATRi and CHK1i, and WEE1i. The combination of CHK1i and ATRi was found to synergise in vivo and in vitro [53,54], apparently through the inhibition of ATR signaling triggered by CHK1i [54]. This suggests the combination is only likely to be useful in small percentage of CHK1i hypersensitive tumours. WEE1i and CHK1i combination have been reported effective [55,56], through a mechanism involving mitotic catastrophe [57]. The combination of WEE1i and CHK1i was most effective in cell lines already highly sensitive to CHK1i as a single agent [26]. Combination treatment with WEE1i and ATRi enhanced replication stress and DNA damage during replication, whereas normal tissue including hematopoietic progenitors and other fast proliferating normal tissue tolerated this treatment [58].

## 5. Chk1i in Combination with Immunotherapy

Recent progress in immunotherapy, particularly immune checkpoint inhibitors (ICI) has demonstrated that engaging the patient’s immune system can increase the duration of clinical response to treatment. An ideal targeted therapy should not adversely affect the patient’s ability to mount an immune response and should trigger pro-inflammatory signals from the tumour to boost immune detection. Small molecule targeted drugs can promote tumour death at all metastatic sites, thus exposing the immune system to a full repertoire of the patient’s tumour antigens. Together these features are likely to enhance the efficacy, and possibly increase the proportion and duration of clinical responses to ICI or other immunotherapies. CHK1i possess many of the properties that would promote a pro-inflammatory response and enhance immune recognition of the treated tumours. CHK1 activated in response to replication stress inhibits the MRE11-MUS81 nucleases, CHK1i relieves this inhibition resulting in extensive DNA damage [59]. In prostate cancer cells, MUS81 is responsible for cytoplasmic DNA that activates the cGAS-STING pathway responsible for interferon induction that can recruit myeloid cells to control tumour growth [60]. Thus, by releasing the inhibition of MUS81, CHK1i increases the level of cytoplasmic ssDNA and cGAS-STING pathway-dependent cytokine production. DNA damage can also trigger increased expression of PD-L1, which is effectively blocked by CHK1i [61] (Figure 4).

CHK1i can also positively influence immune cell function. Activated CD4+ and CD8+ T cells have elevated levels of endogenous DNA damage, a consequence of the very rapid cell cycles triggered by T cell activation [62]. Inhibiting the DNA damage response (DDR) by ATR, which activates CHK1, triggers a durable anti-tumour immune response when used in combination with radiation [63]. This is associated with reduced immunosuppressive regulatory T cell (TReg) numbers and activity, and reduced levels of PD-1 on activated T cells [63]. Combination of CHK1i with low dose of HU that promotes moderate replication stress is well tolerated by all hematopoietic linages in mice, whereas high dose HU in combination with CHK1i completely ablates white cells and severely reduces platelets in blood [19]. Recently, CHK1i has been shown to trigger cGAS-STING cytokine expression and an immune response that is enhanced with anti-PD-L1 in models of small cell lung cancer [64], demonstrating that CHK1i inhibitor based therapy is permissive for immune cell function and can trigger a tumour-directed immune response.

## 6. Conclusions

Exploiting replication stress that is commonly elevated in cancers by inhibiting key regulators of the stress response mechanism is a promising strategy for tumour selective treatment. However, data from recent clinical trials raises two important questions that need to be addressed: (i) What is the best drug combination? and (ii) Do these drugs need to be used at their maximum dose to be effective? The combination of CHK1i with replication stress inducers have been shown to be effective in vitro, but in clinical trials of combination therapy with CHK1i and gemcitabine were highly toxic in patients. The use of the drugs at monotherapy doses that produce high level replication stress was likely to be responsible for this outcome. This activates the ATR-CHK1-WEE1 dependent S phase checkpoint in both tumour and normal proliferating cells. An alternative strategy is to use low doses of replication stress inducer to promote CHK1-dependent dormant fork firing. When used in combination with CHK1i, this appears to have similar anti-tumour efficacy to combination with high replication stress doses but is considerably better tolerated by normal tissue. It is unclear that ATRi will also synergise with this moderate replication stress approach but WEE1i are unlikely to do so. The choice of replication stress inducer may also be an important consideration, as the irreversible effect of drugs such as gemcitabine that make them such effective single agents may impede their ability to produce moderate replication stress. The modest effects on normal tissue viability and proliferation, particularly haemopoietic cells, provide the opportunity to use this combination approach with immunotherapy to potentially extend effective tumour control in the patients.

## Figures and Tables

**Figure 1 cancers-11-01320-f001:**
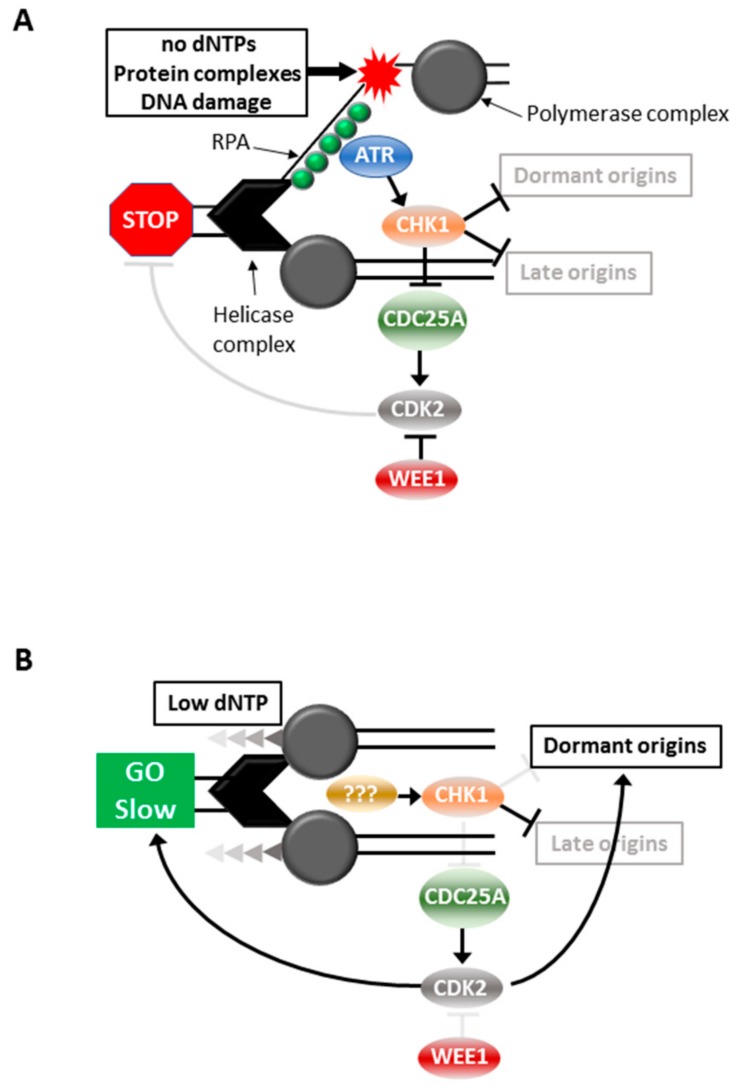
Responses to replication stress. Cell cycle responses to high (**A**) and moderate (**B**) replication stress.

**Figure 2 cancers-11-01320-f002:**
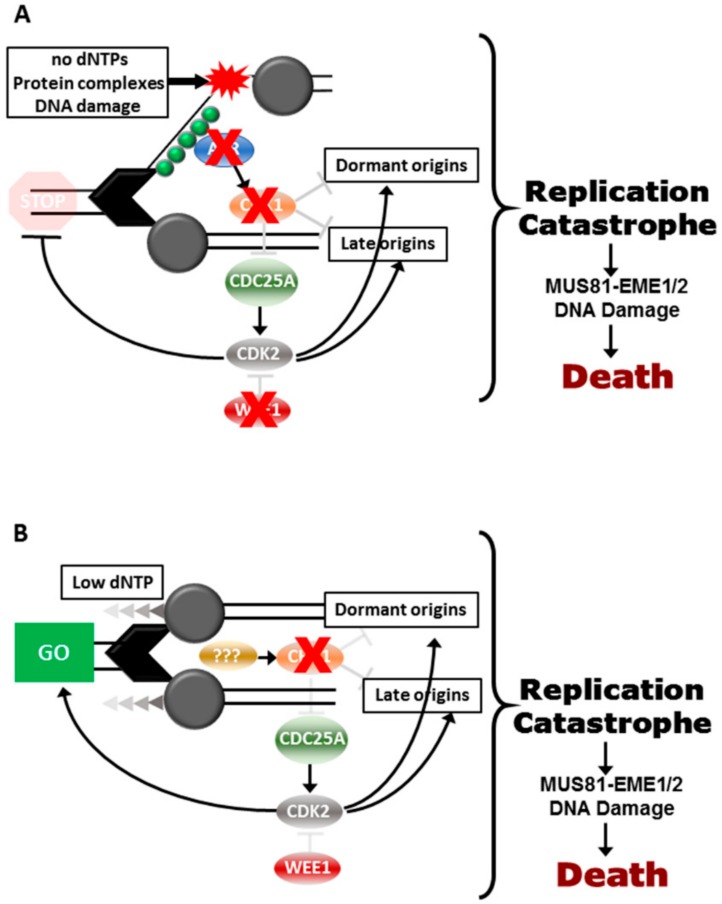
Responses to replication stress. Effects of inhibition of (**A**) ATR, CHK1 or WEE1 in cells with high replications tress, or (**B**) to inhibition of CHK1 in cells with moderate replication stress.

**Figure 3 cancers-11-01320-f003:**
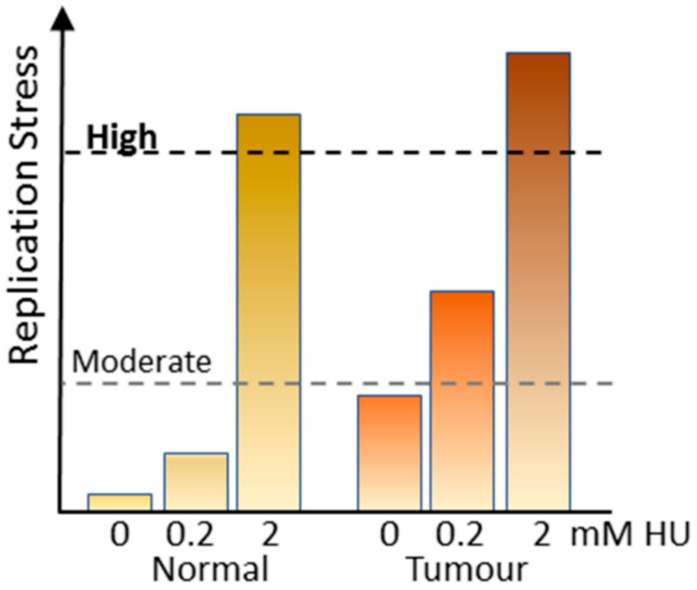
Effects of replication stress on normal and tumour cells. Normal cells have very low levels of replication stress and replication stress imposed using 0.2 mM HU only increases this moderately. Inhibition of CHK1 does not increase this to a point where it has significant toxicity. Tumour cells commonly have elevated replication stress and a moderate stress inducer increases this to a point where CHK1 inhibition can promote excessive origin firing, DNA damage and death. Treatment with high level replication stress inducers does not discriminate between normal and tumour.

**Figure 4 cancers-11-01320-f004:**
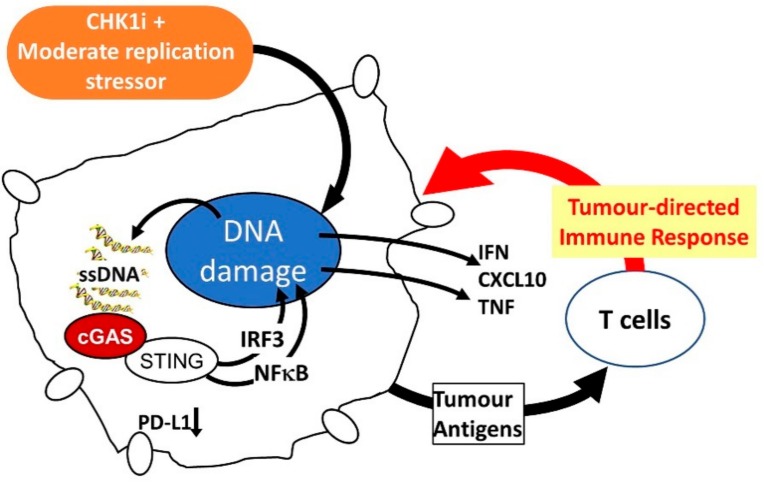
Effects of CHK1i + moderate replication stress inducer on tumour cells to promote an immune response to the tumour.

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
