# Peer review of "Everything in Moderation: Lessons Learned by Exploiting Moderate Replication Stress in Cancer"

_cancers, 2019, doi:10.3390/cancers11091320_

Round 1

Reviewer 1 Report

This short review makes interesting suggestions on the potential benefit of combining inhibitors of DNA damage checkpoint with moderate DNA damage-inducing drugs for cancer treatment. It is well written and clearly expresses the soundness of the proposal. The only shortfall I see is the lack of practical suggestions on combinations of drugs and doses.

The authors should carefullty check the manuscript for typos, repetitions and clarity. In particular at lines 98, 112, 171, 178, 185, 187, 221.

Author Response

We thank the reviewer for their positive response to the review and have attended to the typo and specific line items noted in their review.  

This short review makes interesting suggestions on the potential benefit of combining inhibitors of DNA damage checkpoint with moderate DNA damage-inducing drugs for cancer treatment. It is well written and clearly expresses the soundness of the proposal. The only shortfall I see is the lack of practical suggestions on combinations of drugs and doses.

We have not added in any suggestions as requested as we believe that further research is required, especially in terms of the concentrations of drugs.  We have however, provided guidelines for the characteristics of the drugs and their effects on S phase progression in the text.

The authors should carefully check the manuscript for typos, repetitions and clarity. In particular at lines 98, 112, 171, 178, 185, 187, 221.

We have carefully proof-read the manuscript for typos etc as suggested, and rectified the specific lines identified by the reviewer. 

Reviewer 2 Report

This review compiles the results from various clinical studies from DDR pathway kinase inhibitors, for chemotherapy and provides perspectives on how the existing combination therapy can be improved. The review provides quite comprehensive information about CHK1 inhibitors, in particular, and what other chemotherapeutic agents it may work best with to target cancer cells while minimizing toxicity to normal cells. 

The premise of this review is important as there are different types of replication stresses that can be targeted and thus, it is important to understand the exact mechanistic point where each compound can be used with maximum effectiveness. The authors have done a good review of the primary replication inhibitors used for chemotherapy and give sound advise on where CHK1i are likely to fit in best with.

I think this review can be a good resource for the scientists and clinicians, both, and recommend this as acceptable for publication. My only minor comment is that in line 187, there seems to be a typing error that makes the sentence confusing: "...normal and tumor cells such as RPA are..."

Please replace RPA with the cell lines that the authors would like to mention here.

Author Response

We thank you for your positive response to the review, and have rectified the specific points you have highlighted.

This review compiles the results from various clinical studies from DDR pathway kinase inhibitors, for chemotherapy and provides perspectives on how the existing combination therapy can be improved. The review provides quite comprehensive information about CHK1 inhibitors, in particular, and what other chemotherapeutic agents it may work best with to target cancer cells while minimizing toxicity to normal cells. 

The premise of this review is important as there are different types of replication stresses that can be targeted and thus, it is important to understand the exact mechanistic point where each compound can be used with maximum effectiveness. The authors have done a good review of the primary replication inhibitors used for chemotherapy and give sound advise on where CHK1i are likely to fit in best with.

I think this review can be a good resource for the scientists and clinicians, both, and recommend this as acceptable for publication. My only minor comment is that in line 187, there seems to be a typing error that makes the sentence confusing: "...normal and tumor cells such as RPA are..."

Please replace RPA with the cell lines that the authors would like to mention here.

We have attended to the specific comment on line 187 and altered the text to clarify the point being made. 

Reviewer 3 Report

  This manuscript is designated to review how inhibition of cellular responses to replication stress are potentially exploited for chemotherapeutic purposes, particular focusing on ATR/CHK1/WEE1-dependent pathways which are differently activated depending on the level of replication stress. authors start this topic with describing mechanisms in which replication stresses result from various endogenous and exogenous insults, then addressing high and moderate replication stresses trigger distinct cellular responses. On the basis of this knowledge, they discuss potency of inhibitors of ATR/CHK1/WEE1 for the combined chemotherapy with conventional anticancer agents to enhance replication stress. In this context, it is mentioned that previous clinical approaches have not been successful in term of selective targeting to cancer because of considerable toxic effects to non-cancer cells; however, the combination CHK1i with limited dose of replication stress inducers is effective to cancer cells but causes minimal toxicity to non-cancer cells. Given this information, authors suggest that the synergistic effect of CHK1 inhibitor with the lower dose of anti-cancer agents can be exploited to obtain maximum anti-tumor activity with minimal normal tissue toxicity.

   In my opinion, this manuscript is logically structured and the discussion regarding application of the CHK1 inhibitor for chemotherapy is sound. I believe that this review will be of interest for not only researchers but also clinicians in the relevant fields. However, some sentences/phrases were found to be obscure or unclear and thus I suggest the authors to make improvement in some part of context before acceptance of the manuscript. I listed specific points below. In addition, this manuscript will gain from further proofreading.

In many parts, authors simply state that, in checkpoint-deficient cells, failure of replication or repair results in ‘DNA damage’ (e.g. ‘CHK1i treatment produces ~DNA damage’ [line 171], ‘inhibition of CHK1 ~ results in DNA damage’ [line 186], [line 139]). I think that under-replication or fork collapse generates not only ‘DNA damage’ (double strand breaks, etc.) but also directly causes problems in mitosis or aberrant chromosome structures. I suggest the authors to add more mechanistic aspects to explain the event as the consequence of replication failure.

Line 173-174 ‘moderate replication stress produces much lower levels of normal tissue toxicity in vitro and in vivo~’: these experiments appear to be very important in the topic of this review. Please add appropriate citation(s) at the end of this sentence.  

Line 46: Is ‘ready identification’ supposed be ‘ready and identification’?

Line 98: Is ‘the mechanism of this’ supposed be ‘this mechanism’?

Line 98: CHK1-mediateddegradation; no space between words.

Linde 183-187 ‘However, ~DNA damage observed’: I think this sentence is too long.

Line 205 ‘with is likely~’: Is this supposed to be ‘it is likely’

Line 208-209 ‘incorporation of ~action’: This part is unclear to me.

Line 226 ‘The combination with CHK1i’: What is combined with ‘CHK1i’?

Author Response

We thank the reviewer for their positive response to the manuscript.  We have carefully proof-read the revised manuscript and rectified the specific points raised in your review.  We have not added the extra mechanistic details requested as there are several excellent recent reviews of this particular area of research, and this detail is not central to our review. 

 This manuscript is designated to review how inhibition of cellular responses to replication stress are potentially exploited for chemotherapeutic purposes, particular focusing on ATR/CHK1/WEE1-dependent pathways which are differently activated depending on the level of replication stress. authors start this topic with describing mechanisms in which replication stresses result from various endogenous and exogenous insults, then addressing high and moderate replication stresses trigger distinct cellular responses. On the basis of this knowledge, they discuss potency of inhibitors of ATR/CHK1/WEE1 for the combined chemotherapy with conventional anticancer agents to enhance replication stress. In this context, it is mentioned that previous clinical approaches have not been successful in term of selective targeting to cancer because of considerable toxic effects to non-cancer cells; however, the combination CHK1i with limited dose of replication stress inducers is effective to cancer cells but causes minimal toxicity to non-cancer cells. Given this information, authors suggest that the synergistic effect of CHK1 inhibitor with the lower dose of anti-cancer agents can be exploited to obtain maximum anti-tumor activity with minimal normal tissue toxicity.

   In my opinion, this manuscript is logically structured and the discussion regarding application of the CHK1 inhibitor for chemotherapy is sound. I believe that this review will be of interest for not only researchers but also clinicians in the relevant fields. However, some sentences/phrases were found to be obscure or unclear and thus I suggest the authors to make improvement in some part of context before acceptance of the manuscript. I listed specific points below. In addition, this manuscript will gain from further proofreading.

 We have carefully proof-read the manuscript for typos and logic flow as suggested, and rectified the specific lines identified by the reviewer. 

In many parts, authors simply state that, in checkpoint-deficient cells, failure of replication or repair results in ‘DNA damage’ (e.g. ‘CHK1i treatment produces ~DNA damage’ [line 171], ‘inhibition of CHK1 ~ results in DNA damage’ [line 186], [line 139]). I think that under-replication or fork collapse generates not only ‘DNA damage’ (double strand breaks, etc.) but also directly causes problems in mitosis or aberrant chromosome structures. I suggest the authors to add more mechanistic aspects to explain the event as the consequence of replication failure.

We have not added these extra mechanistic details as there are several excellent recent reviews on this area that we have referenced, and this detail is not central to this review.

Line 173-174 ‘moderate replication stress produces much lower levels of normal tissue toxicity in vitro and in vivo~’: these experiments appear to be very important in the topic of this review. Please add appropriate citation(s) at the end of this sentence.  Done

 Line 46: Is ‘ready identification’ supposed be ‘ready and identification’? Done

 Line 98: Is ‘the mechanism of this’ supposed be ‘this mechanism’? Done

 Line 98: CHK1-mediateddegradation; no space between words. Done

 Linde 183-187 ‘However, ~DNA damage observed’: I think this sentence is too long. Done

 Line 205 ‘with is likely~’: Is this supposed to be ‘it is likely’ Done

Line 208-209 ‘incorporation of ~action’: This part is unclear to me. Done

Line 226 ‘The combination with CHK1i’: What is combined with ‘CHK1i’? Done

Reviewer 4 Report

This is an interesting review that will be of interest to researchers across several disciplines with a focus on cancer biology and the development of cancer treatments.

The section of most interest in this review is the description of the interaction of Chk1i with immnotherapy though this section could be improved with a better description of how this synergy results in enhanced tumour cell killing. A figure/diagram of these interactions would help with the description?

Minor typographical errors:

29- typo: gemcidabine

98 missing space: mediateddegredation

178-224 Numerous examples of extra spaces between words, missing words and extra words that render some sentences nonsensical.

The manuscript as a whole will be improved with additional proofreading

Author Response

We thank the reviewer fro their positive response to our manuscript.  We have attended to the specific points raised in your review.  

This is an interesting review that will be of interest to researchers across several disciplines with a focus on cancer biology and the development of cancer treatments.

The section of most interest in this review is the description of the interaction of Chk1i with immnotherapy though this section could be improved with a better description of how this synergy results in enhanced tumour cell killing. A figure/diagram of these interactions would help with the description?

I have included a new figure to illustrate the effects of CHK1i on tumours cells that are likely to trigger or enhance an immune response. 

Minor typographical errors:

29- typo: gemcidabine

98 missing space: mediateddegredation

178-224 Numerous examples of extra spaces between words, missing words and extra words that render some sentences nonsensical.

The manuscript as a whole will be improved with additional proofreading

We have corrected the specific errors identified above and reproofed the manuscript